# Multi-Personalities Guided Deep Monte Carlo Search for Complex Card Games: A Guandan Case Study

## Abstract

Human personality-enabled AI has Prevailed in various areas, such as complex card games. In these complex multi-agent interaction scenarios, players' decisions are often significantly affected by personality and strategy style, making it difficult for traditional deep Monte Carlo methods to meet practical needs. In this paper, we propose a multi-personality guided deep Monte Carlo search framework, in which three rule-based personalities are incorporated as priors to bias the policy search. Experimental results show that the framework performs well in the Guandan game, which can quickly adapt to the rule prior and gradually discover better strategies through training. This study provides an effective solution for personalized decision-making in complex card games and multi-agent systems, and opens up a new direction for incorporating human style into deep reinforcement learning.

## 1 Introduction

Monte Carlo (MC) methods have long been employed in complex decision-making tasksKemmerling et al. (2024), utilizing large-scale sampling to evaluate action sequences and identify those with the highest estimated expected returns. When combined with deep neural networks for state representation and policy/value estimation, this approach evolves into DMC, which has achieved superhuman performance in perfect information games such as Go Silver et al. (2016; 2017), Chess Silver et al. (2018), and Shogi Schrittwieser et al. (2020).

Despite these successes, DMC methods are often computationally intensive due to their reliance on extensive simulations. Furthermore, their statistically driven decision-making process lacks the capacity to capture the underlying intent or stylistic diversity of human actions, limiting their expressiveness in domains where personalized behavior and interpretability are critical, such as imperfect information games and multi-agent systems.

Guandan, a popular multiplayer card game in China, exemplifies the challenges of imperfect information games due to its large branching factor, team-based coordination, and intricate rule system. As shown in Figure 1, Guandan features larger state and action spaces compared to other representative card games, posing additional difficulties for conventional MCTS-based approaches Shen et al. (2020) and deep reinforcement learning methods Pan et al. (2022); Yanggong et al. (2024). In addition to its structural complexity, the game also involves rich strategic diversity: Human players often demonstrate distinct play styles, which may be aggressive, conservative, or balanced. However, existing DMC-based models are generally incapable of capturing such personalized behavioral patterns, thereby limiting their applicability in realistic, human-aligned gameplay settings. The detailed rules of Guandan are provided in Appendix A.

To address these challenges, we propose a multi-personality guided deep Monte Carlo tree search framework. Our approach integrates human-like play styles by combining rule-based priors for exploration with a dual-head network architecture: a Personality-head that generates action probabilities for state-action pairs based on different personalities, and a Q-head that estimates the value of state-action pairs during exploitation through DMC.

The main contributions of this work are:

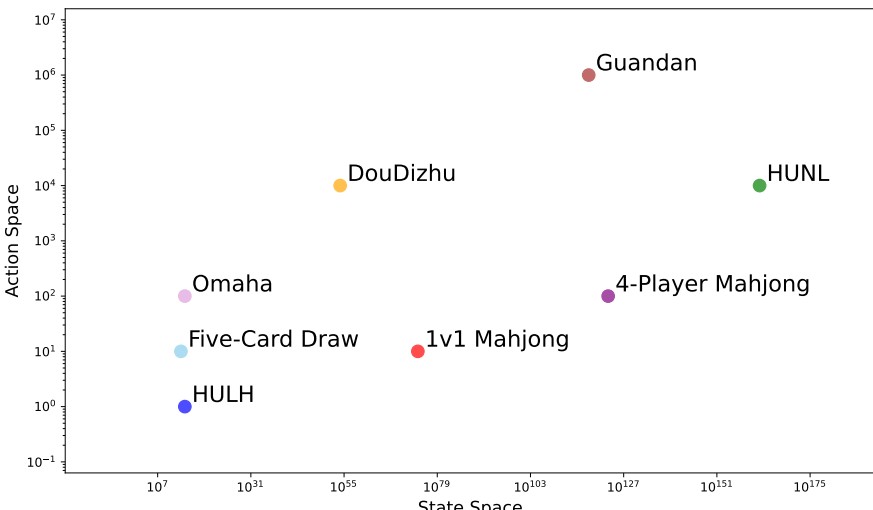

Figure 1: The space of action and state in several representative card games, including Heads-Up Limit Texas Hold'em (HULH), Heads-Up No-Limit Texas Hold'em(HUNL), Omaha Poker, Five-Card Draw, 1v1 Mahjong, 4-Player Mahjong, DouDizhu , and GuanDan.

- **Multi-personality guided DMC framework:** We introduce a novel framework that leverages rule-based priors for guided exploration, improving decision-making and strategy adaptation in complex environments.
- **Controllable action style during deployment:** The proposed framework supports dynamic adjustment between personality-driven and value-maximizing behaviors during action selection, enhancing strategy flexibility and interpretability.
- **Exploration of personality-guided decision-making:** We investigate the influence of personality-driven strategies on gameplay in Guandan, providing valuable insights into the integration of human-like decision-making in AI systems for strategic games.

As far as we know, we are the first to introduce a player AI method that incorporates personality-driven decision-making into a DMC system, enabling it to learn and simulate different styles of game strategies.

## 2    RELATED WORK

### 2.1    RL AND MCTS IN GAME AI

Reinforcement learning (RL) has made significant progress in game AIPinto & Coutinho (2018); Silva et al. (2018); Perez-Liebana et al. (2019). But Imperfect-information games introduce hidden information and randomness, posing greater challenges. Traditional approaches such as Counterfactual Regret Minimization (CFR)Neller & Lanctot (2013) and abstraction techniquesBowling et al. (2017); Brown et al. (2019) have been widely used in poker. Advanced systems like Deep-StackMoravčík et al. (2017) and LibratusBrown & Sandholm (2019) improve strategic optimization and hidden information modelingHeinrich & Silver (2016); Brown & Sandholm (2018), while Deep Q-Networks (DQN)Mnih et al. (2015) have extended RL success to domains like StarCraftVinyals et al. (2019), DOTABerner et al. (2019), HanabiLerer et al. (2020), Honor of KingsYe et al. (2020), and No-Press DiplomacyGray et al. (2020).

In multiplayer card games, RL has been applied to Mahjong, Texas Hold'em, and DoudizhuYou et al. (2020); Zhang et al. (2021); Luo & Tan (2023). Frameworks like DouZeroZha et al. (2021) and DanZeroLu et al. (2023) address large state-action spaces without handcrafted abstractions. DouZero+Zhao et al. (2022) enhances opponent modeling and strategy learning, while Delta-DouJiang et al. (2019) achieves expert-level play via self-play and strategic reasoning. Despite

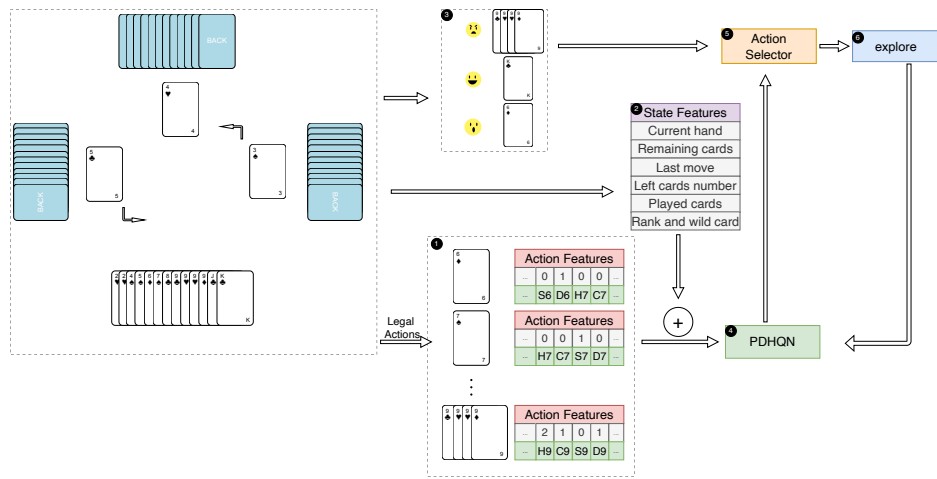

Figure 2: Personality-guided Deep Monte Carlo Search Process

these advancements, limitations remain in pre-trained model dependenciesZhao et al. (2024) and the absence of personalized behavior modeling, highlighting opportunities for further improvement.

## 2.2 IMITATION LEARNING

Inverse Reinforcement Learning (IRL) Ng et al. (2000); Abbeel & Ng (2004); Ziebart et al. (2008) infers reward functions from expert demonstrations, while Generative Adversarial Imitation Learning (GAIL) Ho & Ermon (2016) aligns agent behavior with expert trajectories through adversarial training. Behavioral Cloning from Observation (BCO) Torabi et al. (2018) further expands IL by enabling imitation from state-only demonstrations.

In contrast, we incorporate personality priors into the DMC framework. Rather than relying on demonstrations, our method uses high-level personality traits to guide exploration and elicit diverse, interpretable play styles.

## 3 METHOD

In this section, we detail the overall structure of the proposed Personality-aware Dual-Head Q-Network (PDHQN) from four aspects: feature extraction, personality simulation, model structure, exploration and utilization. The general process is illustrated in the Figure 2.

### 3.1 FEATURE EXTRACTION

In the design of this method, PDHQN needs to intake state-action pairs to generate corresponding Q values and personality distributions. Therefore, how to extract relevant information becomes a crucial issue.

For action features, we construct an action as a 54-dimensional vector, where each dimension represents the suit and points corresponding to the poker card. Since Guandan uses two sets of poker cards, each dimension may take values of 0, 1, and 2. A specific encoding example is shown ❶ in Figure 2.

For state features, shown by ❷ in Figure 2, we encode all the observable information of the current player, including several 54-dimensional vectors to represent the current hand, remaining hand (after excluding played cards), the last card played, and teammate's last card. We also use three 28-dimensional one-hot vectors to encode the number of remaining cards of each opponent, three 54-dimensional vectors to track cards played by each of the other players, three 13-dimensional vectors to encode the current level, opponent's level, and our level, and a 12-dimensional vector to indicate the presence of wild cards.

---

**Algorithm 1** Personality-Guided Rule-Based Action Selection

---

**Input**: Current hand $H$, legal actions $A$, personality type $P$
**Output**: Selected action $a^*$
 1: Adjust the card-type value function $V_P$ based on personality $P$
 2: Segment hand $H$ into candidate combinations $\mathcal{C} = \{C_1, C_2, ..., C_n\}$ that approximately maximize hand value under $V_P$
 3: **for** each $C_i \in \mathcal{C}$ **do**
 4:     Compute $v_i = V_P(C_i)$
 5: **end for**
 6: Select $C^* = \arg\max_{C_i \in \mathcal{C}} v_i$
 7: Derive corresponding action $a^*$ from $C^*$
 8: **return** $a^*$

---

### 3.2 PERSONALITY SIMULATION

The ❸ of Figure 2 illustrates the personality simulation module, which guides action selection using heuristic priors based on predefined rules to emulate distinct human-like play styles. This design is intended to support training stability, enhance the interpretability of agent behavior, and guide exploration with minimal overhead. It is based on an improved version of a framework from the award-winning solution in the First China AI Competition.

We define three personality types: Aggressive, Conservative, and Balanced, each influencing decisions through customized card value functions. As shown in Algorithm 1, the agent:

- Aggressive: Prioritizes strong combinations like bombs and straights.
- Conservative: Emphasizes safer types such as singles and small pairs.
- Balanced: Retains default values without adjustment.

For a comprehensive description of the personality-driven rule-based card-playing algorithm, including detailed parameter settings and value adjustments, please refer to Appendix B.

### 3.3 MODEL STRUCTURE

We adopt a Personality-aware Dual-Head Q-Network (PDHQN) architecture, which consists of a shared feature encoder, a Q-value estimation head, and a personality classification head, as shown in Figure 3. The shared encoder takes a concatenated state-action pair as input and processes it through three fully connected layers with 512 hidden units and `tanh` activations. It extracts a joint representation that is fed into both the Q-value head and the personality classification head for subsequent prediction tasks.

For Q-value prediction, the encoded features are passed through two additional fully connected layers (512 units, `tanh`) followed by a single linear output node, which estimates the expected return of the given state-action pair. In parallel, the personality classification head processes the same shared features via two hidden layers (1024 and 512 units, using `ReLU` activations), and outputs a four-dimensional sigmoid vector. Each dimension corresponds to a probability score for a predefined personality label: aggressive, conservative, balanced, or exploration.

### 3.4 EXPLORATION WITH PERSONALITY-GUIDED ACTION SELECTION

In the personality-guided deep Monte Carlo tree search, exploration serves as a critical component of the decision-making process. At each decision point, an action is stochastically selected from two sources: the personality algorithm, which generates candidate actions based on predefined behavioral styles, and the PDHQN model, which predicts Q-values and samples actions using a softmax policy. This design enables the simulation to integrate both heuristic-driven personalization and value-based optimization. The overall process is illustrated in Figure 2.

In each round of the game, the state feature $s$ and all legal action features $A(s)$ are extracted, as shown in ❶ and ❷. These features are concatenated and passed to the PDHQN model in ❹ to com-

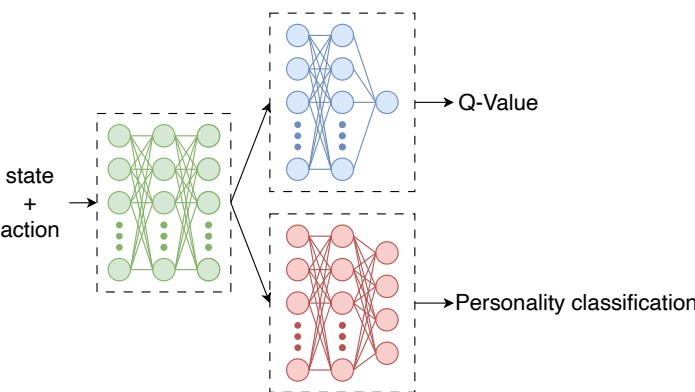

Figure 3: Personality-aware Dual-Head Q-Network

pute Q-values $Q(s, a)$ for all legal actions. An action is then selected according to the probability distribution defined by the softmax equation below and submitted to the action selector in ❺:

$$P(a|s) = \frac{\exp\left(Q(s, a) - \max_{a' \in A(s)} Q(s, a')\right)}{\sum_{a' \in A(s)} \exp\left(Q(s, a') - \max_{a'' \in A(s)} Q(s, a'')\right)} \tag{1}$$

This softmax-based distribution over Q-values retains a preference for high-value actions while introducing stochasticity. Compared to uniform random exploration, it maintains directional guidance toward promising actions, allowing the agent to discover potentially superior strategies beyond the limitations of rule-based heuristics.

The exploration process in ❻ continues until a full round is completed, after which rewards are assigned based on team performance. If the team finishes first, it receives a base reward. Additional points (+3, +2, +1) are given if teammates place second, third, and fourth, respectively. Penalties (-1, -2, -3) apply if teammates rank lower. This reward design promotes teamwork while introducing competitive pressure to enhance multi-agent dynamics.

### 3.5 TRAINING PHASE

During training, we employ a distributed actor-learner framework where multiple actors perform self-play simulations in parallel and a central learner updates the PDHQN model. Each actor runs the Guandan game independently, playing against copies of itself, and transmits gameplay data including state, action, reward, and the personality label of the selected action to the learner after each round.

The personality label is represented as a 4-dimensional one-hot vector: the first three dimensions correspond to the predefined personality heuristics, while the fourth denotes actions chosen through PDHQN-based exploration. These labels serve as the supervision signal for the personality classification head.

The learner updates the PDHQN parameters using the collected samples and periodically synchronizes the updated weights to all actors.

PDHQN features two output heads, each optimized with a distinct loss function. The personality classification head is trained using sigmoid cross-entropy loss:

$$\mathcal{L}_{\text{personality}} = -\frac{1}{N} \sum_{i=1}^{N} \left[ y_i \log(\sigma(z_i)) + (1 - y_i) \log(1 - \sigma(z_i)) \right], \tag{2}$$

where $y_i$ is the ground-truth personality label, $z_i$ is the logit output, and $\sigma(z_i)$ is the sigmoid activation:

$$\sigma(z) = \frac{1}{1 + \exp(-z)}.$$

The Q-value head is trained using mean squared error (MSE) between predicted and target Q-values.

## 3.6 PERSONALITY ACTION SELECTION

To effectively utilize the trained PDHQN model during the deployment phase, we design a composite action selection algorithm that balances value-based optimization with personality-driven behavior.

Assume there are $n$ legal actions and $m$ personality types. For each action, the algorithm computes a combined score by integrating the Q-value and personality classification outputs from the two heads of the PDHQN model. The full process is shown in Algorithm 2.

---

**Algorithm 2** Action Selection with Q-Values and Personality Probabilities

---

**Input**: Q-values $q \in \mathbb{R}^n$, personality matrix $p \in \mathbb{R}^{n \times m}$, blending weight $\alpha$, temperatures $\tau_1, \tau_2$
**Output**: Action indices $a_{\mathrm{idx}} \in \mathbb{R}^m$ for each personality type

1: Normalize Q-values: $q_{\mathrm{norm}} \leftarrow q/(\sum_{i=1}^{n} q_i + \epsilon)$
2: **for** $i = 1$ to $n$ **do**
3:     $p_{\mathrm{row}}[i,:] \leftarrow \mathrm{Softmax}(p[i,:]/\tau_1)$
4: **end for**
5: **for** $j = 1$ to $m$ **do**
6:     $p_{\mathrm{norm}}[:,j] \leftarrow \mathrm{Softmax}(p_{\mathrm{row}}[:,j]/\tau_2)$
7: **end for**
8: **for** $i = 1$ to $n$ **do**
9:     **for** $j = 1$ to $m$ **do**
10:         $s_{i,j} \leftarrow \alpha \cdot q_{\mathrm{norm},i} + (1-\alpha) \cdot p_{\mathrm{norm},i,j}$
11:     **end for**
12: **end for**
13: **for** $j = 1$ to $m$ **do**
14:     $a_{\mathrm{idx},j} \leftarrow \arg\max_i s_{i,j}$
15: **end for**
16: **return** $a_{\mathrm{idx}}$

---

We first normalize $q$ so that its values sum to 1. Then, we apply a two-stage normalization to $p$: a row-wise softmax with temperature $\tau_1$ (where $0 < \tau_1 < 1$) is used to amplify the differences in personality classification across actions under the same state, followed by a column-wise softmax with temperature $\tau_2$ (where $\tau_2 > 1$) to smooth the overall distribution and mitigate bias caused by uneven visitation of state-action pairs during training.

The final score matrix is computed by blending the two components with a weight $\alpha$:

$$s = \alpha \cdot q_{\mathrm{norm}} + (1-\alpha) \cdot p_{\mathrm{norm}}.$$

For each personality type, the action with the highest score is selected:

$$a_{\mathrm{idx}} = \arg\max(s, \mathrm{axis} = 0).$$

Here, each index in $a_{\mathrm{idx}}$ corresponds to the selected action under a specific personality type, where the index ranges from 0 to $m-1$, representing the influence of different personalities on the final action choice. This process enables the agent to exhibit distinct personality traits while still pursuing high-reward actions.

## 4 EXPERIMENTS

Our model training is performed on a computer equipped with an AMD Ryzen 7 5800x 8-core CPU @ 3.80GHz and a GeForce RTX 3090 GPU. The specific training hyperparameters are shown in the Table 1.

All experiments are conducted based on the official open-source Guandan environment released by the First China AI Guandan Competition. This environment provides a standardized simulation platform with well-defined rules and supports reproducible evaluation of agent performance. In

Table 1: Training Hyperparameters

| Hyper-parameters | Meaning | Value |
|---|---|---|
| q lr | Q Value Learning Rate | 0.00025 |
| personality lr | Personality Classification Learning Rate | 0.0005 |
| q optimizer | Q Value Optimizer | Adam |
| personality optimizer | Personality Classification Optimizer | Adam |
| actor num | Number of actors | 8 |
| lamda | Range of clip Q | 0.65 |
| pool size | Memory pool size in learner | 65536 |
| batch size | The batch size for training | 8192 |
| training freq | How many receptions of data between each training | 200 |

order to ensure the reliability and stability of the results, 10 complete games were arranged for each test. Each game will have an average of about 100 rounds, resulting in thousands of card actions. Therefore, we believe that the experimental scale is sufficient to accurately evaluate the accuracy of model actions and reasonably reflect the winning rate of the model under different game conditions.

## 4.1 BASELINE SELECTION

We design two categories of experiments: performance evaluation and personality-conditioned action consistency accuracy evaluation, each with distinct baseline settings.

For performance evaluation, we adopt the state-of-the-art Danzero architecture Lu et al. (2023) as the baseline. In addition to the original $\epsilon$-greedy exploration strategy, we also test a softmax action selection variant to compare different exploration mechanisms under the same DMC framework.

For personality-conditioned evaluation, we use a random policy agent as the baseline, which uniformly samples legal actions. This simple baseline helps isolate the behavioral effects of personality guidance in PDHQN and, by generating diverse scenarios, enables a more robust assessment of personality-aligned action consistency.

## 4.2 PERFORMANCE EVALUATION

In this experiment, we analyze the learning efficiency of Q-value estimation by fixing the personality-weighting factor $\alpha = 1$, thereby isolating the Q-value head and excluding the effects of personality-guided decision-making.

To benchmark performance, we quantify the win rates of PDHQN against two baseline agents constructed on the Danzero architecture: one employing $\epsilon$-greedy exploration with $\epsilon = 0.01$, and the other utilizing a softmax action selection strategy. Both baseline agents are trained for 7,500 iterations to ensure stable and competitive performance, and serve as fixed opponents throughout the evaluation.

Table 2 reports the win rates of PDHQN against two fixed Danzero-based baselines and a random action opponent at various training iterations, illustrating that PDHQN achieves competitive performance under limited training.

Table 2: Win rates (%) at different training stages against baseline opponents (each trained for 7,500 iterations) and a random action opponent.

| Training Iterations | 10 | 50 | 100 | 200 | 400 |
|---|---|---|---|---|---|
| Ours vs. $\epsilon$-greedy | 31.78 | 48.67 | 66.15 | 44.83 | 59.48 |
| Ours vs. softmax | 45.35 | 53.38 | 62.99 | 53.38 | 53.26 |
| Ours vs. random | 62.96 | 74.77 | 69.81 | 92.65 | 98.21 |

Table 3 presents the winning rates between different rule-based strategies. The results indicate a clear strength hierarchy: Balanced ¿ Aggressive ¿ Conservative. Specifically, the balanced strategy

Table 3: Rule-based algorithm battle winning rate. The win rate of each row is achieved by test in the first column against other.

|  | Aggressive | Conservative |
|---|---|---|
| Balanced | 96.77% | 77.33% |
| Conservative | 44.76% | - |

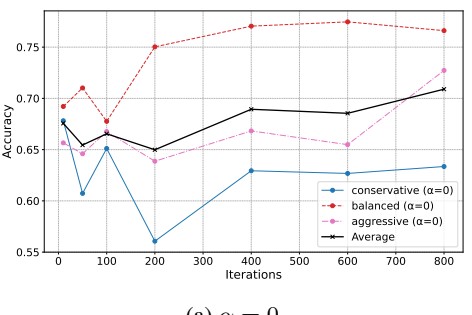

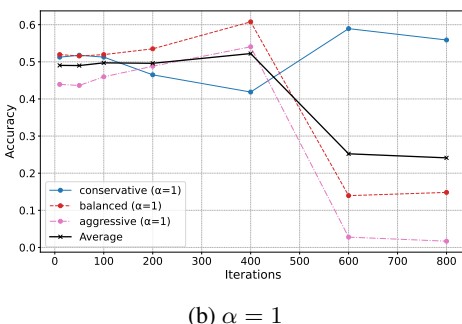

(a) $\alpha = 0$          (b) $\alpha = 1$

Figure 4: Action type accuracies under different epochs when playing against the random card-playing algorithm.

demonstrates a significantly higher winning rate against both aggressive (96.77%) and conservative (77.33%).

### 4.3 ACCURACY EVALUATION

We evaluate the model's performance using *action type accuracy*, which measures whether the type of action selected by the model aligns with that produced by a rule-based algorithm. Note that the same semantic action (e.g., bomb, straight, or pair) may correspond to different action indices due to suit variations. Therefore, we compare action types rather than raw action indices.

The action type accuracy is defined as:

$$A_{\text{accuracy}} = \frac{1}{N} \sum_{i=1}^{N} \mathbb{I}(A_m^i = A_r^i)$$

Where $N$ is the total number of actions, $A_m^i$ is the action type selected by the model for the $i$-th action, $A_r^i$ is the action type derived by the rule-based algorithm for the $i$-th action, and $\mathbb{I}(A_m^i = A_r^i)$ is 1 if the actions match, and 0 otherwise.

We compare the effect of different $\alpha$ values on action type accuracy in Figure 4. When $\alpha = 0$ (a), the policy relies solely on the personality head, without incorporating Q-value information. In this case, the accuracy consistently improves over training iterations and reflects the strength of the personality prior: stronger rule-driven personalities yield higher accuracy. When $\alpha = 1$ (b), the policy depends solely on the Q-value head, where accuracy initially rises but drops sharply after 400 iterations, indicating a shift from personality priors to alternative high-return strategies.

In Table 4, we report action type accuracy at 400 and 800 iterations under different $\alpha$ values and personalities against a random baseline. At 400 iterations, rule-based priors yield higher accuracy, while by 800 iterations the model increasingly departs from these priors to explore new strategies.

### 4.4 RESULTS ANALYSIS

**Rule strength and learning deviation:** When a personality rule does not consistently lead to optimal outcomes, the Q-value head tends to explore alternative strategies as training progresses. This

Table 4: Action type accuracy results under different $\alpha$ and personalities with random card-playing algorithm for 400 and 800 iterations

| $\alpha$ | 400 Iterations | | | 800 Iterations | | |
|---|---|---|---|---|---|---|
| | Aggressive | Conservative | Balanced | Aggressive | Conservative | Balanced |
| 0 | 66.82% | 62.93% | 77.04% | 72.72% | 63.35% | 76.60% |
| 0.1 | 66.12% | 60.51% | 75.91% | 68.37% | 59.38% | 60.96% |
| 0.2 | 67.91% | 54.68% | 76.08% | 69.92% | 56.21% | 59.02% |
| 0.3 | 67.90% | 60.97% | 76.26% | 66.10% | 59.00% | 59.88% |
| 0.4 | 66.29% | 63.28% | 74.57% | 63.29% | 56.37% | 57.21% |
| 0.5 | 68.05% | 51.54% | 75.39% | 67.90% | 51.27% | 58.82% |
| 0.6 | 69.38% | 64.53% | 75.32% | 66.06% | 56.44% | 57.67% |
| 0.7 | 70.71% | 61.33% | 69.29% | 64.41% | 59.31% | 55.69% |
| 0.8 | 70.51% | 64.69% | 71.07% | 67.04% | 57.59% | 59.83% |
| 0.9 | 67.51% | 61.69% | 69.83% | 64.51% | 56.05% | 55.14% |
| 1 | 54.04% | 41.85% | 60.76% | 1.68% | 55.87% | 14.82% |

results in a noticeable decline in action type accuracy, especially under the *Aggressive* and *Balanced* personalities, indicating a shift away from rule-based behavior.

**Personality head learning stability:** The personality head maintains a steady upward trend in learning performance, suggesting that it successfully captures and preserves personality-aligned action preferences. This separation ensures that the model retains distinct behavioral traits even when Q-value strategies diverge from the initial rules.

**Effect of priors in early training:** Personality-driven priors provide valuable guidance in the early training phase, helping to stabilize learning. However, as training progresses, the model increasingly prioritizes strategies that outperform the initial rules, reflecting a natural transition from imitation to optimization.

## 5 CONCLUSION

In this study, we proposed a personality-guided deep Monte Carlo Tree Search framework that incorporates human-like play styles by combining rule-based priors. The network architecture decouples value estimation and personality alignment through two separate output heads: a Q-value head for reward prediction and a personality head for capturing stylistic preferences.

Experimental results demonstrate that personality priors improve exploration efficiency. As training progresses, the model gradually transitions from rule-based behavior to discovering novel strategies beyond the initial personality priors. The $\alpha$-weighted decision mechanism enhances both interpretability and adaptability of agent behavior.

**Generalizability** The proposed framework is generalizable and can be extended to other multi-agent, imperfect-information environments such as various card games, by modifying the feature extraction process to fit the domain.

**Limitation** Due to computational constraints, the number of training actors and iterations was limited, restricting full exploration of the state space. More extensive training resources could unlock richer personality dynamics and more refined strategic behaviors.

**Future Work** Future directions include incorporating opponent and teammate modeling to improve coordination in multi-agent settings. Such enhancements would strengthen the agent's adaptability, strategic flexibility, and social reasoning, extending the impact of personality-aware decision-making in complex environments.

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

# A    Rule of Guandan

Guandan uses two decks of cards (108 cards, including Jokers). Four players form two teams, with teammates seated opposite each other. Each player is dealt 27 cards, referred to as a "hand." The game proceeds in a counterclockwise order. The first player in each round leads the trick and can play any valid card combination. Other players must either follow with a higher card combination of the same type, play a Bomb, or pass. A trick ends when three players pass consecutively, and the last player to play cards leads the next trick.

A round ends when all players except one have emptied their hands, or both players of the same camp finish their hands. Players are ranked based on the order in which they finish, as Banker (1st), Follower (2nd), Third (3rd), and Dweller (4th).

## A.1    Card Combinations

In Guandan, card combinations include standard and special types. Standard combinations consist of Solo (a single card), Pair (two cards of the same rank), Triple, Tube (three consecutive Pairs), Plate (two consecutive Triples), Full House (a Triple combined with a Pair), and Straight (five or more consecutive cards, where Aces can serve as 1). Special combinations include Bombs (four or more cards of the same rank), Flush Straights (five cards of the same suit in consecutive rank, capable of beating Bombs with fewer than six cards), and Joker Bombs (two Red Jokers and two Black Jokers, which can beat any other card type). Bombs can beat any standard combination, with their strength determined first by the number of cards and then by their rank. Flush Straights can beat smaller Bombs, while Joker Bombs override all other combinations.

## A.2    Tribute and Level Cards

From the second round onward, the Dweller of the previous round must tribute their highest single card (excluding wild cards) to the Banker. The Banker must return a card with a rank no higher than 10. If there are Double-Dwellers, each must tribute their highest card, and the Banker accepts the higher-ranked tribute. If the tribute-paying player possesses two Red Jokers, the tribute phase is skipped, and the Banker leads the first trick. Cards matching the current level are known as Level Cards, which rank just below Jokers when played singly but can function at their natural rank when forming combinations. Level Cards in the Hearts suit act as wild cards, substituting for any needed card in combinations, except Jokers.

The objective of Guandan is to level up from 2 to A, with teams advancing based on their rankings at the end of a round. If the Banker and the Follower belong to the same team, the team advances three levels. If the Banker and the Third belong to the same team, the team advances two levels. If the Banker and the Dweller belong to the same team, the team advances one level. Opponents of a Double-Dweller automatically advance their level by three. When the level reaches Q or K and the winning team is eligible to advance by two or three levels, Level A cannot be skipped. At Level A, a team can only win if the Banker's partner finishes as either the Follower or the Third.

## A.3    Winning Conditions

A Guandan game consists of multiple rounds. The first team to surpass Level A wins the game. If the game has a fixed number of rounds, the team at the highest level when the rounds conclude is declared the winner. In the event of a tie, an additional round is played to determine the victor. If a team at Level A fails to win three consecutive rounds, they are demoted back to Level 2. This ensures a competitive balance and adds strategic depth to the gameplay.

# B    Personality-driven Rule-based Card-playing Algorithm

We propose a simple yet effective rule-based algorithm that adapts card-playing strategies according to the player's hand, personality, and game situation to select the most appropriate action.

Table 5: Action Value Scores Based on Type and Rank

| Type / Rank | 2-7 | 8 | 9 | T | J | Q | K | A | curRank | B | R |
|---|---|---|---|---|---|---|---|---|---|---|---|
| Single | | | | | -1.0 | | | | -0.5 | 0 | 0.5 |
| Pair | | | -1.0 | | | | -0.5 | 0 | 0 | 0.5 | 1.0 |
| Trips / ThreeWithTwo | | -1.0 | | -0.5 | -0.5 | -0.5 | 0 | 0 | 0.5 | 0.5 | 1.0 |
| TwoTrips / ThreePair | | | | | | -0.5 | | | | | |
| Straight | -1.0 | -0.5 | -0.5 | -0.5 | -1.0 | -1.0 | -1.0 | -1.0 | -1.0 | -1.0 | -1.0 |
| Bomb | | | | | | 1.0 | | | | | |
| StraightFlush | | | | | | 1.0 | | | | | |

Table 6: Action Value Adjustments by Personality Type

| Action Type Card Type | Free Action RV Adjustment | | Restricted Action RV Adjustment | |
|---|---|---|---|---|
| | Aggressive | Conservative | Aggressive | Conservative |
| Single | -1.0 | +1.0 | - | - |
| Pair | -1.0 | +1.0 | - | - |
| Trips | -1.0 | +0.5 | - | - |
| ThreeWithTwo | - | +0.5 | - | - |
| ThreePair | +1.0 | -1.0 | - | - |
| TwoTrips | +1.0 | - | - | - |
| Straight | +1.0 | -0.5 | - | - |
| Bomb | +1.5 | -1.0 | +1.5 | -1.5 |
| StraightFlush | +1.0 | -1.0 | +1.0 | -1.0 |

## B.1 EXPLANATION OF THE ALGORITHM

The pseudocode describes a personality-guided decision-making process for selecting optimal actions in card play. The key steps are:

- **Initialization:** The game state is initialized via `SetBeginning`, setting the player's position, card counts, and resetting all strategy-related variables.
- **Action Generation and Evaluation:** The algorithm first partitions the hand into possible actions by maximizing intrinsic card values (see Table 5), then applies personality-based adjustments to these candidate actions to compute their final scores.
- **Personality-Driven Strategy Adjustment:** `makeReviseValues` adjusts revised values (`freeActionRV`, `restrictedActionRV`) based on personality:
  - **Aggressive:** Prioritizes high-impact actions such as Bomb or Straight.
  - **Conservative:** Favors safe actions like Single or Pair.
  - **Balanced:** Leaves revised values unchanged, acting as a neutral policy.

  Adjustment weights are detailed in Table 6.
- **Dynamic Updates:** `UpdateRV` functions adjust revised values based on game context. For example, when a partner has only one card left, the value of Single increases to support coordination; conversely, when an opponent has one card, Single is penalized to avoid enabling their win.
- **Optimal Action Selection:** The action with the highest combined value (base + adjusted) is selected as the optimal move.

## B.2 EXAMPLE SCENARIO

Given the hand `['S7', 'H7', 'C7', 'D7', 'CA', 'CA', 'S5', 'D5']`, the system first partitions the hand by maximizing the cumulative base action value, without applying any personality-related adjustments. The partition that maximizes the base score yields a total value of **0**, with the following candidate actions:

- Bomb: ['S7', 'H7', 'C7', 'D7'] (base value = 1)

- Pair: ['CA', 'CA'] (base value = 0)

- Pair: ['S5', 'D5'] (base value = -1)

**Personality-Specific Decisions:**

Personality-driven adjustments (see Table 6) are then applied to compute the final adjusted value for each action. The system selects the action with the highest adjusted score under each personality profile:

- **Aggressive:**
    - Bomb: $1 + 1.5 = 2.5$
    - Pair (A): $0 - 1.0 = -1.0$
    - Pair (5): $-1 - 1.0 = -2.0$

    The highest adjusted value is **2.5** for Bomb, so the system selects the Bomb ['S7', 'H7', 'C7', 'D7'].

- **Conservative:**
    - Bomb: $1 - 1.0 = 0.0$
    - Pair (A): $0 + 1.0 = 1.0$
    - Pair (5): $-1 + 1.0 = 0.0$

    The highest adjusted value is **1.0** for Pair (A), so the system selects the Pair ['CA', 'CA'].

- **Balanced:**
    - No adjustments applied; base values are used directly.

    The highest value is **1.0** for Bomb, so the system selects the Bomb ['S7', 'H7', 'C7', 'D7'].

