# OpenReview forum: "Multi-Personalities Guided Deep Monte Carlo Search for Complex Card Games: A Guandan Case Study"
_ICLR.cc/2026/Conference — Submitted to ICLR 2026_

### Official Review · Reviewer_FvBH · 2025-10-29

**Soundness:** 2
**Presentation:** 2
**Contribution:** 2
**Rating:** 2
**Confidence:** 5

**Summary:**

The paper introduces a novel multi-personality guided deep Monte Carlo tree search (DMC) framework designed to integrate human-like play styles into AI decision-making for complex card games, using the Chinese game "Guandan" as a case study. The authors propose a dual-head network architecture (PDHQN) that separates Q-value estimation and personality alignment. The framework incorporates three predefined personality types—aggressive, conservative, and balanced—to guide exploration and improve interpretability. Experimental results demonstrate that the framework achieves competitive performance and can adapt beyond rule-based strategies as training progresses.

**Strengths:**

1. The paper explicitly incorporates personality-driven decision-making into a DMC framework, providing a new perspective on integrating human-like styles into strategic AI systems.

2. Dual-Head Network Design: The separation of Q-value estimation and personality classification ensures interpretability while allowing the model to balance between value optimization and stylistic preferences.

**Weaknesses:**

1. One major limitation is the lack of evaluation against human players, which has been done in two-player Fighting Games [1]. The authors focus solely on AI-versus-AI experiments, leaving unanswered questions about whether the three predefined personality types (aggressive, conservative, balanced) adequately represent the diversity of human strategies. Human players often exhibit more nuanced and context-dependent behaviors that may not be fully captured by these rigid personality definitions. For instance, a human player might switch between aggressive and conservative play depending on the game state, which the current framework does not explicitly address. The part can

2. The paper defines personality types through heuristic rules (e.g., aggressive prioritizes bombs and straights), but these definitions seem overly simplistic. Human strategies are often more fluid and situational, blending elements of different styles. For example, in Guandan, a player might adopt a conservative strategy early in the game to preserve strong cards but switch to aggressive play later to secure a win. This adaptability is not reflected in the current framework.

3. The authors do not provide a clear evaluation of how well the AI aligns with human expectations or whether it can mimic human-like gameplay. Metrics like "human-likeness" or subjective evaluations from human players would strengthen the claims about personality-driven decision-making.

4. The framework uses only three personality types, which might limit its flexibility. Human players may adopt hybrid or entirely different styles that do not fit neatly into these categories. Expanding the personality space or allowing for dynamic personality blending could improve the model's realism.

5. The experimental evaluation was insufficient, and no comparison was made with the current baseline of reinforcement learning algorithms in Guandan.

ref:

[1]https://openreview.net/forum?id=eN1T7I7OpZ

**Questions:**

1. Could the author provide a complete stylistic assessment? In particular, what are the differences between this assessment and the actual strategic styles of human players?

2. How did the author consider switching between different styles during a match?

3. There's another core question the authors need to address: in real AI-human matches, is a stylized AI truly necessary for a game like Guandan? This is particularly important to discuss, as it determines the necessity of this work. Should stylized AI be a bonus in the game AI training process, or is it a separate issue requiring further discussion?

4. Could you provide a comparison with other baselines, like baselines in [1] or some baselines in Doudizhu? Including win rate and stylistic differences?

ref:

[1]https://ieeexplore.ieee.org/abstract/document/10584299

---

### Official Review · Reviewer_jHhY · 2025-10-31

**Soundness:** 2
**Presentation:** 2
**Contribution:** 2
**Rating:** 2
**Confidence:** 4

**Summary:**

This paper introduces a method for developing multi-personality-guided Guandan agents. The Q-networks are enhanced with an additional personality classification head to adjust action selection based on specific personalities. The proposed algorithm outperforms three baseline methods in terms of win rate and is also compared with a rule-based personality-guided algorithm.

**Strengths:**

This paper investigates a relatively new and challenging game domain. The topic of multi-personality agents is less commonly explored, as the mainstream research tends to focus solely on maximizing win rates.

**Weaknesses:**

In my opinion, the definitions of the three personality types are not sufficiently elaborated or formalized. I am also unclear about the rationale behind incorporating different personalities in Guandan strategies. For example, the description of the aggressive personality—"Prioritizes strong combinations like bombs and straights"—strikes me as a naturally suboptimal strategy that could harm win rates, and I doubt experienced players would adopt such an approach. The win rates of the aggressive agent against balanced and conservative agents are not clearly presented.

I find the main evaluation method for personality adherence—using accuracy based on a rule-based policy as ground truth—somewhat unconvincing.

I also notice a lack of references to existing work on personality-aware game agents. While I am not deeply familiar with this topic, I found at least one related reference [1], and I believe there are likely more. Additionally, though not explicitly modeling personality, reference [2] involves learning agents with varying levels of aggressiveness.

[1] Holmgård, Christoffer, et al. "Monte-carlo tree search for persona based player modeling." *Proceedings of the AAAI Conference on Artificial Intelligence and Interactive Digital Entertainment*. Vol. 11. No. 5. 2015.

[2] Shen, Ruimin, et al. "Generating behavior-diverse game ais with evolutionary multi-objective deep reinforcement learning." *Proceedings of the Twenty-Ninth International Conference on International Joint Conferences on Artificial Intelligence*. 2021.

**Minor Issues:**

- The citation style appears unconventional to me. Author names are expected inside the brackets, and some references seem to lack spacing.
- I find Table 3 somewhat difficult to read. It might be clearer to structure it with three columns corresponding to the three possible algorithm pairs.
- Line 377 contains unexpected "¿" symbols.

**Questions:**

1. In Table 2, why does the win rate against $\epsilon$-greedy and softmax agents decrease after 100 iterations?
2. Why were the baselines trained for 7500 iterations, while PDHQN was trained for significantly fewer?
3. Does the policy genuinely adheres to the predefined personality? Can the authors demonstrate, for example, that the aggressive agent behaves more aggressively than the balanced agent?
4. What is the personality matrix in Algorithm 2?
5. What are the advantages of the proposed algorithm compared to the rule-based method?

---

### Official Review · Reviewer_j2eZ · 2025-11-01

**Soundness:** 2
**Presentation:** 2
**Contribution:** 2
**Rating:** 2
**Confidence:** 4

**Summary:**

This paper introduces an approach to guiding a Monte Carlo search-based player for Guandan with rule-based personalities. The approach works by separately learning Q-networks for traditional q-values and personality values and then guides action selection based on balancing these. They present experiments comparing to Danzero, a pure RL approach from 2023. They find that their approach is comparable to Danzero and can be fairly controllable in terms of personalities depending on training time.

**Strengths:**

In terms of originality, the paper takes a fairly unique approach to personality-guided MCTS, though there are prior personality-based MCTS work that go uncited [1].

In terms of quality, the extra analysis around personality accuracy is of high quality, even as the base play experiment is of somewhat lower quality.

The paper overall has high clarity. The basic approach is well explained as is the experimental setup.

The significance is relatively low. It is really only relevant to those interested in automated game playing, and even then it is not state of the art for Guandan and there is prior personality-based MCTS work.

1. Holmgård, Christoffer, et al. "Automated playtesting with procedural personas through MCTS with evolved heuristics." IEEE Transactions on Games 11.4 (2018): 352-362.

**Weaknesses:**

The paper has a number of weaknesses that could be addressed to improve the paper.

The first is that some of the claims in the introduction are not supported. The claim "improving decision-making and strategy adaptation in complex environments" is not supported, as decision-making is not clearly improved and strategy adaptation is inconsistent. The claim about "interpretability" is also not supported as there's no evaluation of the explainability or interpretability. The authors also make a claim around "human-like decision-making", which is not supported. There's no evidence presented that the approach leads to more human-like decision making.

The second is the literature review. The literature review doesn't cover prior personality-based automated game playing work [2,3] and specifically not prior personality-based MCTS work [1]. There also isn't coverage of work on automated Guandan playing since 2023 [4,5]. Both would be necessary to cover given the nature of the work.

The third is the clarity of the figures. Figure 2 and figure 4 have text that is too small to read without zooming into the figure. Similarly, Figure 2 is very dense and could be reformatted so the important elements are larger.

The fourth is the experiment and results. The authors claim that Danzero is SOTA, but it's been replaced by Danzero+ since then [4] along with other approaches [5]. Similarly, the results are inconclusive. The performance of the authors' approach peaks at 100 training iterations without explanation. Similarly the personality aspect of the work is not clearly controllable, and there's no explanation for why "conservative" remains so controllable compared to the other personalities. Improved baselines and more in-depth analysis would improve this part of the paper.

Finally, the text could be improved in terms of several formatting and grammar issues. For example, "The ❸ of Figure 2" -> "❸ of Figure 2", "Balanced ¿ Aggressive ¿ Conservative" -> "Balanced > Aggressive > Conservative", and "by test in the first column against other" -> "by testing the approach in the column against the row".

1. Holmgård, Christoffer, et al. "Automated playtesting with procedural personas through MCTS with evolved heuristics." IEEE Transactions on Games 11.4 (2018): 352-362.
2. Ingram, Branden, et al. "Creating diverse play-style-centric agents through behavioural cloning." Proceedings of the AAAI Conference on Artificial Intelligence and Interactive Digital Entertainment. Vol. 19. No. 1. 2023.
3. Halina, Emily, and Matthew Guzdial. "Diversity-based deep reinforcement learning towards multidimensional difficulty for fighting game ai." arXiv preprint arXiv:2211.02759 (2022).
4. Zhao, Youpeng, et al. "Danzero+: Dominating the guandan game through reinforcement learning." IEEE Transactions on Games 16.4 (2024): 914-926.
5. Zhao, Tianchang, et al. "Playing Guandan with Large Language Models Based on Prompt Engineering." 2025 7th International Conference on Artificial Intelligence Technologies and Applications (ICAITA). IEEE, 2025.

**Questions:**

1. Why did the authors use Danzero as the baseline?
2. How do the authors explain the results vs Danzero and with the inconsistent personality results?

---

### Official Review · Reviewer_29We · 2025-11-01

**Soundness:** 2
**Presentation:** 2
**Contribution:** 2
**Rating:** 2
**Confidence:** 3

**Summary:**

This paper introduces PDHQN, a Deep Monte Carlo Tree Search method for the Guandan card game that incorporates personality styles during action selection. PDHQN consists of a network that encodes state-action pairs and outputs an state-action specific Q-value as well as a personality head classifies the state action pair into three distinct personality types.

**Strengths:**

- The paper tackles the card game Guandan, which is a game that is less explored in the research community.

**Weaknesses:**

- Some major aspects of PDHQN are improperly explained/confusing or missing in the paper
   - The authors mentioned that PDHQN is used for Monte Carlo Tree Search, however the use of MCTS in PDHQN is explained at all in the paper. Is there a dynamics model that enables the state transition for tree search? What is the search budget? Do the authors use UCT? Does the tree search occur during data collection during training?
   - How are the personality ground truth labels obtained? Are some of workers collecting personality specific data? This is unclear to me.
- It is similarly unclear to me what is the utility of different personality types. In Table 2 the PDHQN that is used to evaluate against the different baselines does not utilize the personality head (\alpha = 1). Why are the effects of the personality head excluded here?
- In Table 4, why is PDHQN only evaluated against a random baseline?
- The different personalities could be better explained. Is the personality simply providing different value weightage to to different hands? Does it take into account other factors such as the stage of the game, opponents hands etc.?
- Overall, the quality writing of the paper can be improved. The authors should provide better motivation for why the encoding different personalities are useful in the Guandan game and provide clearer explanations of their proposed method.

**Questions:**

See weaknesses

---

### Meta-Review · Area_Chair_coex · 2025-12-07

**Summary:**

This paper proposes a multi-personality–guided deep Monte Carlo search framework for the Guandan card game, implemented through a dual-head network (PDHQN) that outputs both Q-values and personality classifications. Reviewers generally agree that the paper addresses a relatively underexplored game domain and introduces an interesting direction by explicitly incorporating personality into decision-making.

However, all reviewers identify substantial shortcomings in the methodological clarity, motivation for personality modeling, experimental design, baseline selection, and alignment between claims and empirical evidence. The description of how PDHQN integrates with Monte Carlo Tree Search is incomplete, the definition and acquisition of personality labels are insufficiently justified, and the experiments do not convincingly demonstrate controllable or meaningful personality behaviors. The related-work section omits several key references, and the empirical evaluation lacks comparisons with stronger recent baselines.

Given that these concerns are broad, substantial, and consistently raised across all reviews, I agree with the reviewers that the paper does not meet the acceptance criteria in its current form.

**Reviewer Concerns:**

Below I summarize which concerns appear to remain unresolved, based on the final review texts:

Unresolved Concerns (all reviewers agree they remain open)

Insufficient explanation of the MCTS component in PDHQN.
Reviewers still find no clear description of the search procedure, the presence or absence of a dynamics model, search budget, use of UCT, or the role of tree search during training.

Unclear definition, motivation, and ground truth for personality types.
Reviewers continue to question:

how ground-truth personality labels are obtained,

whether the three personalities are meaningful or realistic in Guandan,

whether the heuristic definitions are justified,

and whether “personality accuracy” is an appropriate evaluation metric.

Lack of evidence that personality meaningfully affects behavior.
Reviewers note that personality is disabled in key experiments (e.g., α = 1 in Table 2) and that the paper does not demonstrate clear, consistent behavioral differences among personalities.

Weak or outdated baseline comparisons and insufficient analysis of results.
Concerns remain regarding:

the absence of more recent Guandan baselines,

the unclear training-iteration discrepancy between PDHQN and baselines,

the unexplained performance trends (e.g., peaking at 100 iterations),

and evaluating PDHQN only against a random agent in Table 4.

Claims in the introduction not supported by evidence.
Reviewers consistently state that the paper does not substantiate claims regarding improved decision-making, strategy adaptation, interpretability, or human-like behavior.

Missing coverage of key related work.
Reviewers highlight incomplete citations of prior personality-based MCTS and other personality-driven game agent research, as well as more recent Guandan AI work.

Writing and presentation issues.
All reviewers note issues with figure readability, formatting inconsistencies, and miscoded symbols.

Given the consistency and emphasis of these unresolved issues, the rebuttal does not appear to have addressed reviewers’ primary criticisms.

**Reviewer Scores:**

Based on the final review texts and the nature of the concerns, I believe none of the reviewers would have changed their overall recommendation if they had participated further in the discussion. Their main reservations are structural and substantive—particularly methodological clarity, experimental validity, and insufficient evidence for the core claims—which are unlikely to be resolved without major revision and additional experiments.

Thus, all reviewers would likely maintain a Reject recommendation.

---

### Decision · Program_Chairs · 2026-01-26

Reject